# Active Soil Filter System for Indoor Air Purification in School Classrooms

**DOI:** 10.3390/ijerph192315666

**Published:** 2022-11-25

**Authors:** Sungwan Son, Aya Elkamhawy, Choon-Man Jang

**Affiliations:** 1Department of Environmental Research, Korea Institute of Civil Engineering and Building Technology (KICT), Goyang 10223, Republic of Korea; 2Smart City and Construction Engineering, University of Science & Technology (UST), Daejeon 34113, Republic of Korea

**Keywords:** active soil filter, indoor air purification, particulate matter (PM), air quality, school classroom

## Abstract

An active soil filter system was newly designed and evaluated to obtain a higher removal efficiency of fine particulate matter (PM) for indoor air purification in schools. Unlike passive air purification systems that remove PM using only plant leaves, air purification can be maximized by filtering polluted indoor air directly between the soil particles supporting the plant. The novel system is composed of a composite soil layer and a suction blower that forces outside air into the soil layer. It was found that the air purification performance was improved as the inflow air velocity upstream of the soil is decreased and the soil stacking height increased. The lower the soil moisture, the better the air purification performance. Considering both the classroom environment and the system’s energy consumption, it is recommended that the soil stacking height is 150 mm, the soil inflow air velocity is 2 cm/s, and the relative humidity is 35%. Under these conditions, the air purification efficiency for PM_2.5_ is 41.5%. The indoor air purification system using the soil filter system, along with the currently used plant leaves medium, is expected to improve the indoor air quality in public facilities, such as school classrooms.

## 1. Introduction

Poor classroom air quality not only adversely affects students’ health but also greatly diminishes their learning ability. Among the air pollutants in indoor spaces, particulate matter (PM) with a diameter of 2.5 to 10 μm is known to affect respiratory health and increase mortality rates after long-term exposure [1,2,3,4,5]. In general, PM_2.5_ is more dangerous than PM_10_ because it penetrates deeper into the respiratory system, decreasing cardiorespiratory function [6,7,8]. According to the World Health Organization (WHO), almost 91% of the world’s population lives in areas with air pollutant levels that exceed the WHO standards; 5 µg/m^3^ and 15 µg/m^3^ for PM_2.5_ and PM_10_, respectively [9]. Remarkably, a 7-year study of American student achievement found that an increase in PM_2.5_ concentration was associated with a decrease in math and English test scores [10]. In addition, a Spanish study found that PM_2.5_ exposure was associated with reduced cognitive development in children [11,12]. In order to actively reduce indoor pollution caused by PM_2.5_, it is necessary to develop high-functioning air purification technology for multi-use facilities, such as schools [13,14]. It must be noted that lower indoor PM concentration has a positive effect on students’ health and disease prevention.

Recently, vegetation filters using plant leaves to purify polluted air have been considered to reduce the PM concentration of airborne pollutants [9,15,16,17]. Several researchers have proposed plant leaves as a medium to control indoor pollutants, but the reduction efficiency has been very low due to the passive reduction property [15,18]. Meo et al. [19] investigated the effects of a green environment on PM and showed a possible PM reduction by a green environment, such as plant leaves. Qin et al. [1] reported an indirect PM reduction by the ventilation of the tree canopy using vegetation in a park. Chen et al. [4] evaluated the importance of the vegetation species, planting composition, and wind in urban areas, and showed that planting composition was more effective than vegetation species choice for PM concentration reduction. Kończak et al. [6] investigated the PM absorption of roadside vegetation. They found that the roughness of the leaf surface affected PM absorption levels, with vines and shrubs having a higher reduction effect. Viecco et al. [20] and Perini et al. [21] studied the absorption of PM by vegetation species and leaves and showed that species with smaller leaves, higher leaf roughness, and the presence of trichomes have excellent effects on PM absorption. As described above, PM adsorption by passive vegetation leaves has limitations regarding the PM reduction efficiency.

On the other hand, technology for an indoor air purification system with a plant-based bio-filter using a mechanical device has been introduced. Irga et al. [22,23] presented a botanical bio-filter system to improve indoor air quality (IAQ). They showed a removal efficiency of PM_2.5_ of 48.21% with mechanical ventilation using coconut husk as the bio-filter. Pettit et al. [17] also reported a PM reduction using the plant bio-filter system. Air purification systems using bio-filters and vegetation leaves have been partially studied, but few studies have considered operating conditions, such as airflow velocity through the filter. The consumption of mechanical energy used for indoor air purification and ventilation is expected to increase [24,25]. Therefore, it is necessary to introduce a new auxiliary device to achieve the carbon-neutral mandate by reducing the energy needed for operating an air purification system with mechanical ventilation in schools.

In the previous study by Elkamhawy and Jang [26], an air purification system consisting of artificial soil-based vegetation and electrostatic precipitator (ESP) filters was designed and experimentally evaluated. The authors reported the air purification performance of PM_2.5_ according to the airflow rate using the vegetation soil filter. In previous studies, general commercially available soil was employed as the medium. Therefore, there were experimental limitations due to the non-uniformity of the soil composition and moisture. It is known that soil properties, such as bulk density, porosity, water resistance, and moisture, are important for plant growth [27].

In the present study, an active soil filter system is newly designed and evaluated through experimental measurements for use in school classrooms. The system consists of a soil filter having a multi-layered structure, a duct system, and an air suction blower. The blower controls the air volume by adjusting the rotational speed of the motor and circulates the polluted air through the multi-layered soil. Unlike existing passive systems using only vegetation leaves, the active vegetation soil filter uses an air blower to directly force polluted air through the soil. The air purification performance is experimentally evaluated according to the stacking height of the soil layer and the soil moisture. An active soil filter system can be installed in public facilities, such as school classrooms, and is expected to be used as an effective indoor air purifier.

## 2. Materials and Method

### 2.1. School Classroom for Air Purification

The experimental performance of the active soil filter system in a school classroom was evaluated using a full-scale performance test-bed [28]. As shown in Figure 1, the classroom test-bed has the same dimension as the standard Korean elementary school classroom. The standard classroom’s length, width, and height are 8.4 m, 7.2 m, and 2.6 m, respectively. The test-bed was equipped with a mechanical ventilation system that supplied clean air to the classroom through a duct system connected to an air handling unit (AHU). The detailed test-bed structure and air conditioning system were described in a previous paper by Son and Jang [28]. During the active soil filter test, only the soil filter was operated while the windows and doors were closed, and the mechanical ventilation system was stopped.

The ventilation rate in the classroom was determined by the number of students and the dimensions of the classroom. The amount of outdoor air required according to the number of students was determined by Equation (1) [29].
(1)$$Q_{SN}=q_{p}N_{sc}$$
where *Q_SN_*, *q_p_*, and *N_sc_* are the required air volume, the outdoor air volume per student in the classroom, and the number of students in the classroom, respectively. ASHRAE Standard 62.1-2019 recommends a *q_p_* = 5.0 L/student [30]. In 2020, the average number of elementary school students per classroom in Korea was 23.1. Based on Equation (1), an air volume of 415 m^3^/h (CMH) is required to maintain a proper CO_2_ concentration in a standard Korean classroom.

On the other hand, the WHO recently announced enhanced ventilation rate requirements to remove the bio-PM [31]. According to the new WHO guidelines, the minimum required ventilation rate for ordinary workplaces and public spaces is 10 L/s per person, i.e., 36 CMH. Therefore, a standard Korean classroom needs 830 CMH of clean air.

### 2.2. Active Soil Filter

An active soil filter was introduced in the present study to effectively adsorb PM from the indoor space of a school classroom. As shown in Figure 2, the soil filter has a multi-layered structure comprising plant leaves, complex soil, and porous material. The polluted air in the space is first partially passively purified using the leaves of the plant, then it passes through the complex soil, which acts as an active air purification filter. An air suction blower is introduced to force the outside air into the soil. In order to minimize the pressure loss due to the duct connected to the blower, the duct length inside the soil filter system was minimized, and the inner diameter of the duct was maximized. In addition, a cap was installed at the end of the duct, as shown in Figure 2b, to prevent inflow into the duct due to soil falls and to maintain uniform inflow velocity.

Soil specifications used for the soil chamber are shown in Table 1. In order to evaluate the air purification performance according to the stacking height of the composite soil, the height (“h” in Figure 2a) varies from 50 mm to 150 mm.

### 2.3. Experimental Apparatus and Methodology

In the present study, air purification effects were evaluated only for composite soil, excluding plants. To ensure the reliability of the experimental data, the flow rate, pressure, and PM concentration were measured according to international standards. The experimental method and preliminary measurement are as follows.

Figure 3 shows the experimental apparatus of the active soil filter system, which is an open-type test apparatus. It consists of a multi-layered soil filter, measurement sensors (PM, pressure, velocity), a flow control system, and a data acquisition system. The specifications of the sensors and measurement equipment used in the present study are described in Table 2. Every measurement instrument was calibrated and adjusted before the experiment, and the fine PM concentration was collected as the ensemble average of the values measured for thirty minutes under each measuring condition.

### 2.4. Soil Moisture Content

Although it is known that the soil moisture content varies depending on the type of plants, in general, 20–40% soil moisture is appropriate for horticulture. A gravimetric method using a soil drying oven and humidification was used for the quantitative evaluation of soil moisture. The required amount of moisture is determined by the difference between the initial weight and the dry weight, and the water content is determined as follows:(2)$$Moisture\;content=\frac{weight\;of\;wet\;soil-weight\;of\;dry\;soil}{weight\;of\;dry\;soil}\times 100$$

Dry soil is based on soil dried in an oven at 110 °C for 24 h. Based on 100 g of dry soil, the amount of supplied water for 15% and 35% soil moisture content for the present study is 10 g and 24.4 g, respectively. Considering the influence of the ambient temperature and moisture on soil in the indoor living environment and the appropriate soil moisture level for vegetables, the PM removal performance of two types of 15% and 35% soil moisture were selected for evaluation in the present study.

### 2.5. Analytical Method for the Soil Filter

The correlation between the inflow air velocity upstream of the soil filter and the pressure loss due to the soil and duct inside the soil chamber was evaluated as a preliminary experiment. The inflow air velocity was determined after measuring the flow velocity at 4 points in the radial direction, according to the AMCA 210 standard, by installing a Pitot tube inside the circular duct connected to the soil filter, as shown in Figure 4. The Pitot tube was installed at a fully developed flow position, 15D (D: duct inner diameter) downstream from the duct entrance, as shown in Figure 3.

The pressure loss caused by the soil and the duct in the soil chamber was measured by installing a pressure tap downstream of the soil (1) in Figure 3a and downstream of the soil chamber (2), respectively. The inflow air velocity to the soil was controlled by an inverter connected to the blower, which changed the rotational speed of the blower motor. Pressure loss according to the height of the soil, the porous material, and the duct is shown in Figure 5. It was noted that the pressure loss increased as the height of the composite soil increased, regardless of the airflow rate, and the pressure loss was relatively small in the porous material, Hyuga soil.

The PM reduction performance of the soil filter was evaluated by measuring the PM sensor upstream and downstream of the soil filter. The experimental apparatus was installed in the full-scale school environmental performance test-bed, and the standard PM_2.5_ of Arizona dust was set at about 100 µg/m^3^. Measurements were performed for 30 min, and a PM_2.5_ concentration of 60 µg/m^3^ was achieved after stabilizing operations for about 1 h to induce a uniform distribution of PM inside the room.

In particular, the purified air by the soil filter was measured by the air sampling port inside the duct, and the hole diameter was determined so that the three holes installed in the port had a constant suction velocity, as shown in Figure 6. Table 3 shows the hole diameter of the air sampling port according to the inflow air velocity. The measured data was stored in the computer in the data acquisition system.

## 3. Results and Discussion

### 3.1. Pressure Loss of the Soil Due to Soil Moisture

Figure 7 shows a pressure loss at the soil height of 150 mm with respect to the inflow air velocity for two soil moistures: 15% and 35%. It should be noted that only the soil loss is compared in Figure 7, excluding the duct loss in the soil chamber, unlike the pressure loss shown in Figure 5. When water is supplied to the soil, water is stored in the pores that existed between the soil grains. Therefore, as the humidity of the soil increases, the area of pores through which air can pass decreases, and the pressure drop of the active soil filter increases. In the figure, the pressure loss by the soil increases approximately twice as much compared to the soil moisture increase from 15% to 35%, regardless of the inflow air velocity.

### 3.2. PM_2.5_ Reduction Performance with Respect to Soil Height and Inflow Air Velocity

The PM_2.5_ reduction performance was measured by installing the active soil filter in the full-scale school environmental performance test-bed. Before the experiment, the PM_2.5_ concentration to 100 µg/m^3^ using standard PM_2.5_ of Arizona dust was set in the classroom. Circulation fans were introduced to make a uniform PM_2.5_ concentration inside the classroom, and the blower linked with the soil filter was operated to reduce the indoor PM_2.5_ concentration. PM_2.5_ reduction performance by the PM_2.5_ concentration was measured upstream and downstream. The indoor PM_2.5_ concentration was evaluated as 60 µg/m^3^. To evaluate the PM_2.5_ reduction performance by the soil filter, the PM_2.5_ reduction rate is defined as
(3)$$\;PM_{2.5}\;Reduction\;Rate=\frac{PM_{2.5,\;upstream}-PM_{2.5,\;downstream}}{PM_{2.5,upstream}}*100$$
where *PM*_2.5,*upstream*_ and *PM*_2.5,*downstream*_ are the concentration levels of PM_2.5_ measured upstream and downstream of the soil filter, respectively.

Figure 8 shows the PM_2.5_ concentration and reduction rate with the soil height (H) of 150 mm for three inflow air velocities while the soil moisture remained at 15 ± 5%. Relatively larger reductions in PM_2.5_ concentration in the upstream and downstream of the soil filter were observed as the inflow air velocity increased to the soil filter system installed in the closed classroom.

As shown in Figure 9, the average PM_2.5_ reduction rate decreased from 48.7% to 43.2% as the inflow air velocity increased from 2 cm/s to 8 cm/s. Additionally, the fluctuation of real-time PM_2.5_ reduction rate increased from 4.7% to 8.5%, based on the average value, as the inflow air velocity increased. This increase is due to the increased motility and degree of freedom between soil grains as the velocity increases.

Figure 10 shows the PM_2.5_ reduction rate with respect to the inflow air velocity for three soil heights. The reduction rate is determined using Equation (3) and the ensemble averaging 180 data points of PM_2.5_ concentration measured every 30 min upstream and downstream of the soil filter. It is noted that the PM_2.5_ reduction rate decreased linearly as the inflow air velocity increased, regardless of the soil height. Average values of the PM_2.5_ reduction rate for the soil height and the inflow air velocity are summarized in Table 4.

### 3.3. Effects of Soil Moisture on the PM Reduction Performance

For ornamental plants grown in classrooms, light, temperature, moisture, and artificial soil are important variables for plant growth. As for the growth rate of crops, it is known that it is better to manage soil moisture at 40% rather than 20%. In order to analyze the PM_2.5_ reduction performance according to soil moisture, the moisture content of the composite soil was evaluated and compared at 35%, in addition to 15%, as described in the previous section.

Figure 11 shows the PM_2.5_ concentration and reduction rate at the soil height (H) of 150 mm for three inflow air velocities while the soil moisture was kept at 35 ± 5%. Compared to the PM_2.5_ reduction rate of 15% soil moisture in Figure 8, the PM_2.5_ reduction rate is generally lower at 35% soil moisture depending on the inflow air velocity, and the pressure loss increases, as shown in Figure 7, due to the mutual coupling of soil grains with higher soil moisture.

Figure 12 shows the PM_2.5_ reduction rate for the soil height and moisture of 150 mm and 35 ± 5%, respectively. The average PM_2.5_ reduction rate decreases from 41.5% to 19.7% as the inflow air velocity increases from 2 cm/s to 8 cm/s. Based on the average value of the real-time PM_2.5_ reduction rate, the percentage deviation between the maximum and minimum values grows from 6.8% to 20.5% as the inflow air velocity increases. The increase in the percentage deviation of PM_2.5_ reduction rate of 35% compared to 15% soil moisture is relatively large because the average value is low; the absolute deviation is similar.

Figure 13 shows the comparisons of PM_2.5_ reduction rate with respect to the inflow air velocity at the soil height of 150 mm for two soil moistures. It can be seen that the PM_2.5_ reduction rate decreases as the inflow air velocity increases, regardless of the soil moisture, but the PM_2.5_ reduction rate increases as the soil moisture decreases. Considering the healthy growth of ornamental vegetables in school classrooms, PM_2.5_ reduction rates, and energy consumption from air intake blowers, it is recommended to maintain soil moisture at 30–40% and minimize inflow air velocity to 2 cm/s.

### 3.4. Application and Expected Effect of Active Soil Filter in School Classrooms

In order to reduce fine particulate matter (PM_2.5_) in a school classroom, there is an air circulation method that supplies fresh air to the room from the outside and uses an air conditioner filter with MERV 13 or higher. Wu et al. [32] found that even short-term exposure to PM_2.5_ in students was associated with increased inflammation-induced airway issues and lung dysfunction. The Ministry of Education of Korea recommends an average concentration of 15 μg/m^3^ for PM_2.5_ in schools, but the WHO has tightened the standard to 10 μg/m^3^. Even a small amount of PM_2.5_ has an adverse effect on students’ health, so it is desirable to lower the concentration of PM_2.5_ as much as possible. When an active soil filter is applied as an auxiliary device to the classroom air purification system along with the use of the AHU, it can reduce the system’s operating time and save energy. Unlike an AHU, the soil filter operates continuously for 24 h, so it not only continuously lowers the PM_2.5_ concentration but also indirectly controls the indoor humidity during the winter season.

Figure 14 shows the configuration of a classroom with three soil filter systems. According to ASHRAE Standard 62.1-2019, the ventilation volume per person in the school classroom is 23.1 m^3^/h. Considering the 21.3 students per class average, a total ventilation volume of 415 m^3^/h is required.

Table 5 shows the ventilation volume according to the three inflow air velocities described in the present study. When the inflow air velocity is 2 cm/s under 35% soil moisture, the PM_2.5_ reduction rate is more than 41.5%, and the ventilation effect is 22.3% compared to the required ventilation for all students. Since the active soil filter is operated for 24 h, the air change rate per day is equivalent to 5.3 times the required ventilation volume. It is noted that when the inflow air velocity of the active soil filter is increased to 5 cm/s and 8 cm/s, the air exchange rate in the classroom is also increased by 2.5 times and 4 times, respectively, compared to the inflow air velocity of 2 cm/s.

## 4. Conclusions

The paper describes an active soil filter system, which was newly designed and evaluated to obtain a higher removal efficiency of fine particulate matter (PM) for indoor air purification in schools. This novel active filter system uses multi-layered composite soil as a filter to maximize the indoor air purification effects compared to the passive air purification system that uses only plant leaves. To evaluate the PM reduction performance by experimental measurements, the system was installed in a full-scale school classroom environment performance test-bed while changing the soil stacking heights, soil moisture, and the inflow air velocity upstream of the soil. The results are summarized as follows.

Increasing the inflow air velocity through the soil and the stacking height of the multi-layered composite soil in the classroom increases the PM reduction efficiency and pressure loss. It was found that the pressure loss by the soil also increases approximately twice as the soil moisture increases from 15% to 35%, regardless of the inflow air velocity.

The average PM_2.5_ reduction rate decreases from 48.7% to 43.2% for soil moisture of 15% and decreases from 41.5% to 19.7% for soil moisture of 35% as the inflow air velocity increases from 2 cm/s to 8 cm/s. Considering the healthy growth of ornamental vegetables in school classrooms and the energy consumption of a suction blower, it is desirable to maintain 35% soil moisture and minimize the inflow air velocity to 2 cm/s.

When the active soil filter is applied as an auxiliary device of the classroom air purification system along with an AHU, it can reduce the operating time of the AHU and thereby save energy. In addition, the soil filter can operate continuously for 24 h, so it not only continuously lowers the PM_2.5_ concentration inside the classroom but also indirectly controls the indoor humidity during the winter season.

## Figures and Tables

**Figure 1 ijerph-19-15666-f001:**
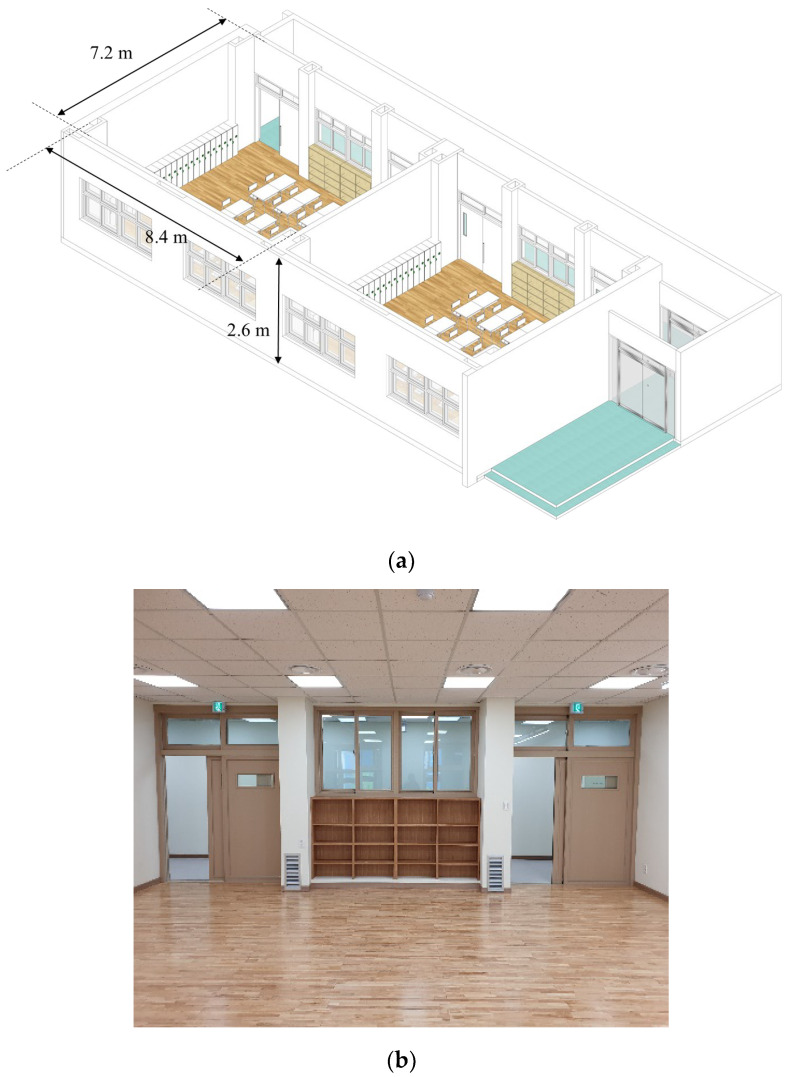
(**a**) Perspective view of the full-scale school environment performance test-bed. (**b**) Picture of a classroom.

**Figure 2 ijerph-19-15666-f002:**
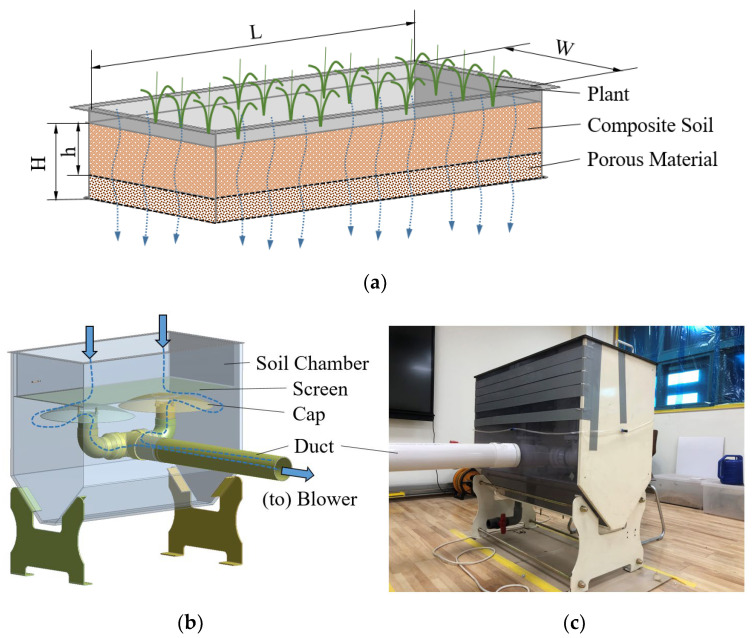
Active soil filter: (**a**) soil chamber; (**b**) perspective view; (**c**) picture. (blue arrows represent flow direction).

**Figure 3 ijerph-19-15666-f003:**
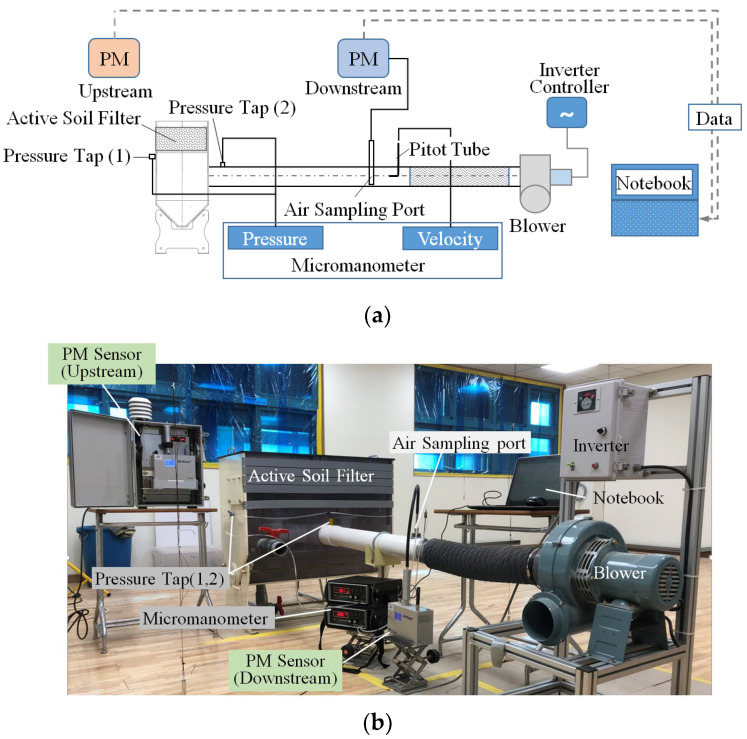
Experimental apparatus of the active soil filter system: (**a**) schematic view; (**b**) picture.

**Figure 4 ijerph-19-15666-f004:**
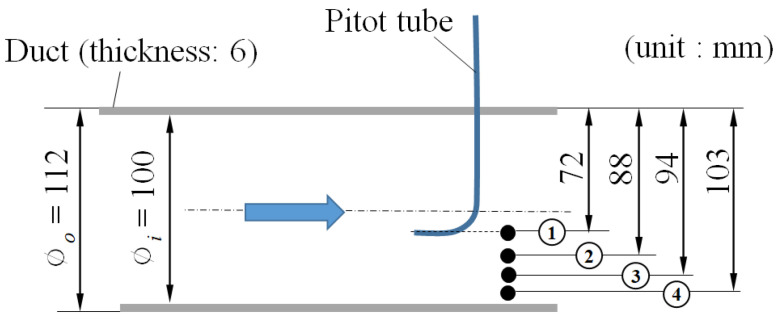
Velocity measuring positions (black dot positions) by a Pitot tube (blue arrow represents flow direction).

**Figure 5 ijerph-19-15666-f005:**
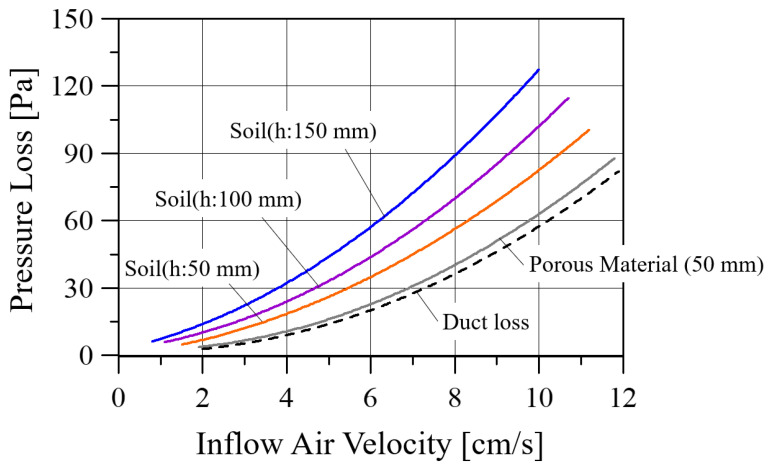
Pressure loss according to the height of the soil, the porous material, and the duct with regards to inflow air velocity.

**Figure 6 ijerph-19-15666-f006:**
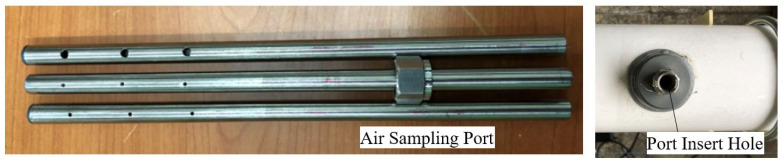
Picture of air sampling ports and port hole.

**Figure 7 ijerph-19-15666-f007:**
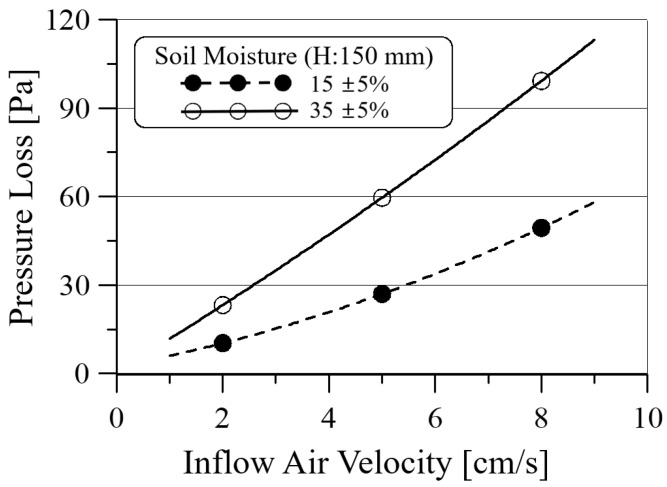
Pressure loss for two soil moisture at the soil height of 150 mm (“H” in Figure 1) with respect to inflow air velocity.

**Figure 8 ijerph-19-15666-f008:**
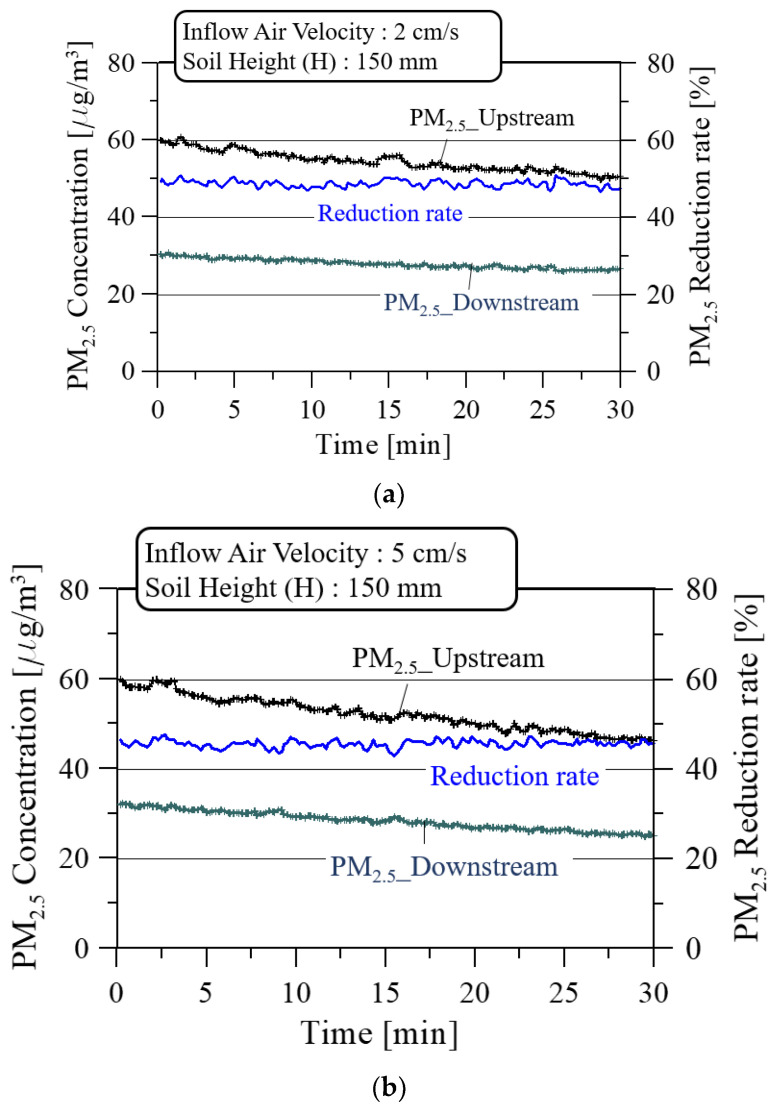
PM_2.5_ concentrations and reduction rate at the soil height (H) of 150 mm and the soil moisture of 15 ± 5% for three inflow air velocities: (**a**) 2 cm/s; (**b**) 5 cm/s; and (**c**) 8 cm/s.

**Figure 9 ijerph-19-15666-f009:**
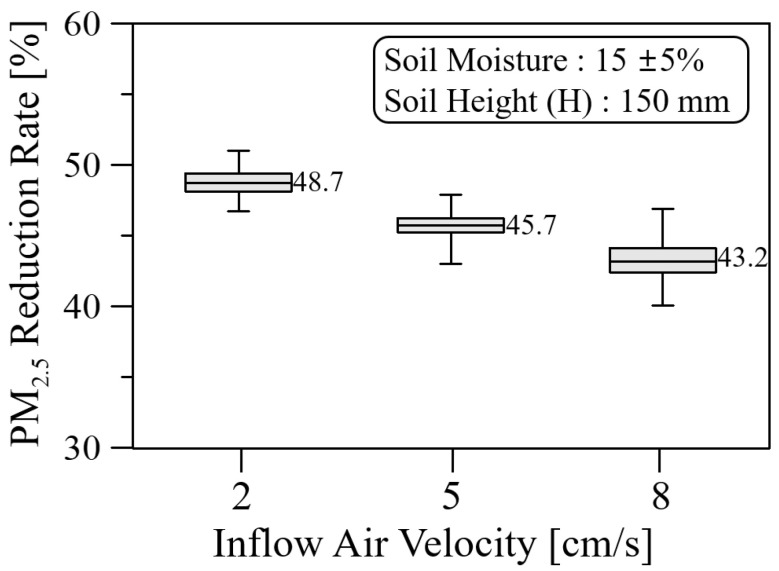
PM_2.5_ reduction rate with respect to inflow air velocity for the soil height and moisture of 150 mm and 15 ± 5%, respectively.

**Figure 10 ijerph-19-15666-f010:**
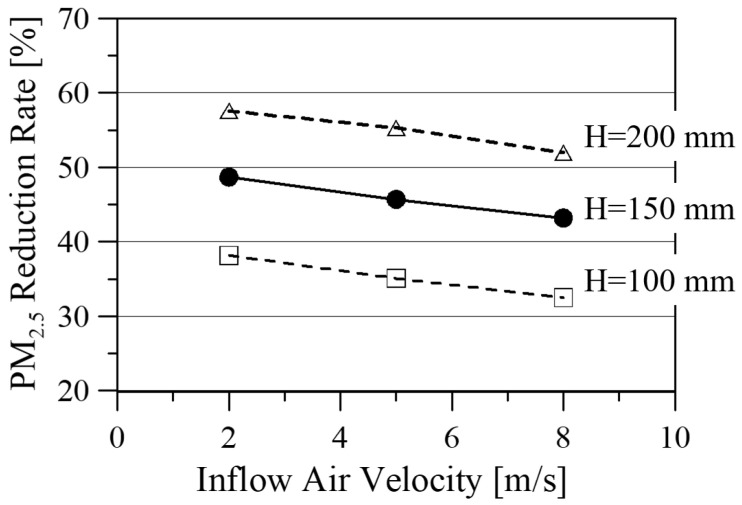
PM_2.5_ reduction rate with respect to inflow air velocity and soil moisture of 15 ± 5%.

**Figure 11 ijerph-19-15666-f011:**
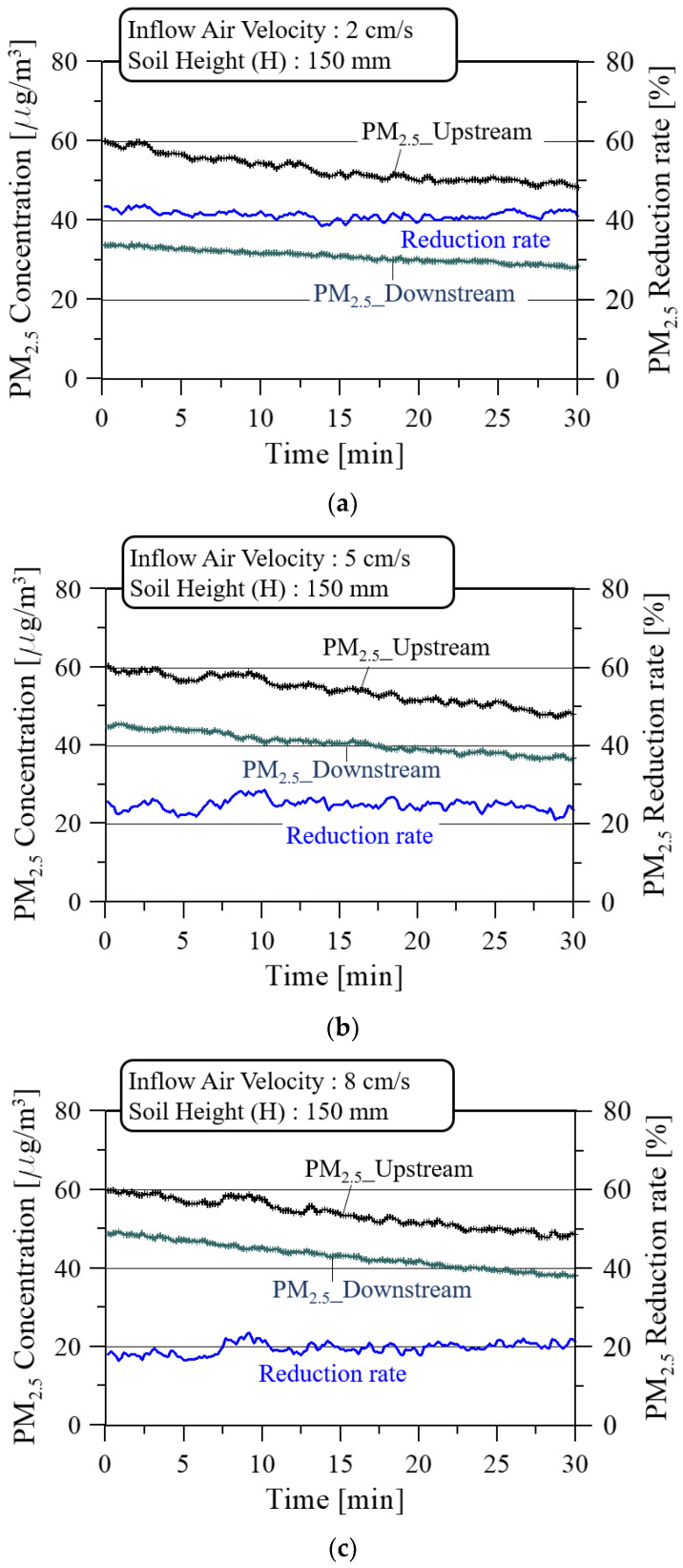
PM_2.5_ concentrations and reduction rate at the soil height (H) of 150 mm and the soil moisture of 35 ± 5% for three inflow air velocities: (**a**) 2 cm/s; (**b**) 5 cm/s; and (**c**) 8 cm/s.

**Figure 12 ijerph-19-15666-f012:**
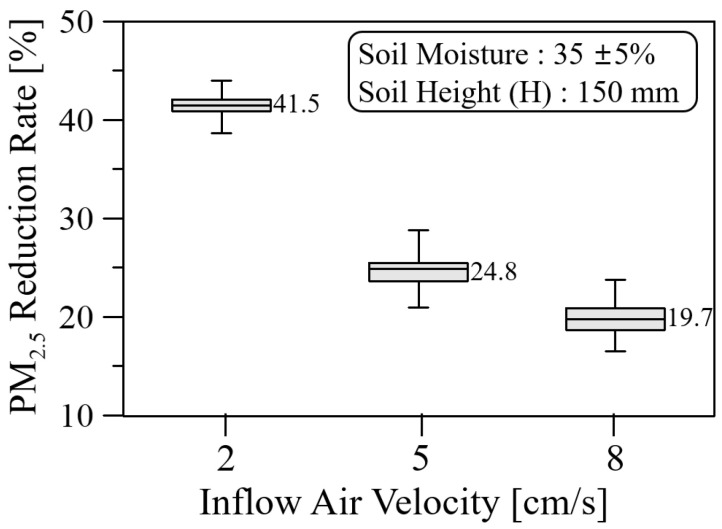
PM_2.5_ reduction rate with respect to inflow air velocity for the soil height and moisture of 150 mm and 35 ± 5%, respectively.

**Figure 13 ijerph-19-15666-f013:**
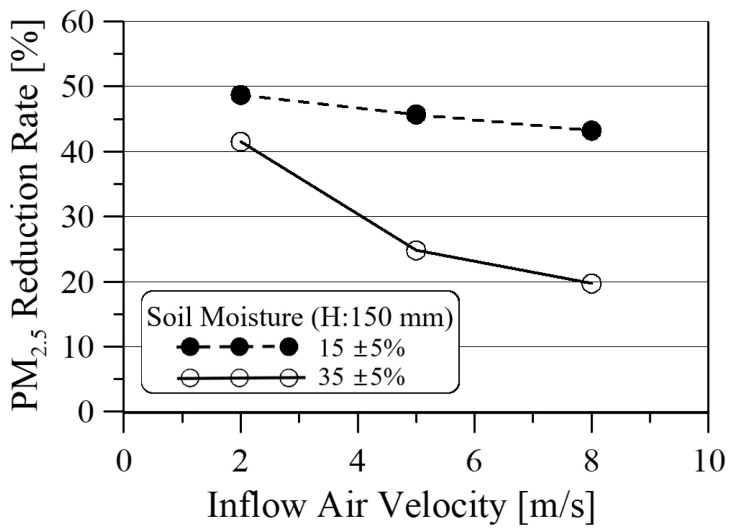
PM_2.5_ reduction rate with respect to inflow air velocity at the soil height of 150 mm.

**Figure 14 ijerph-19-15666-f014:**
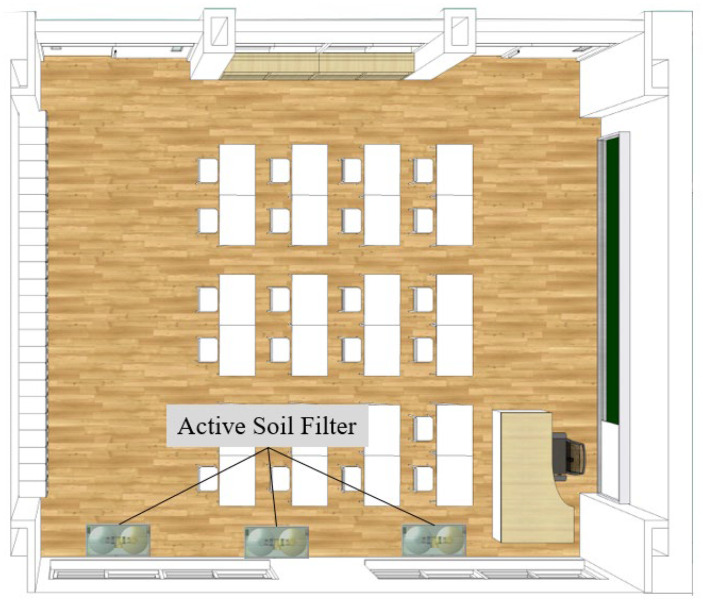
Installation positions of the active soil filter in a classroom.

**Table 1 ijerph-19-15666-t001:** Design specification of an active soil filter.

Parameters	Composite Soil	Porous Material
Height, mm	50, 100, 150	50
Length (L) × Width (W), mm	950 × 450	
Material ingredient	Integrated soil consisting of peat moss (10%), red loam soil (30%), broccoli soil (25%), green soil (25%), and perlite (10%)	Hyuga soil (perlite series)
Grain diameter, mm	1~3	2.5~5
Picture	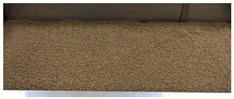	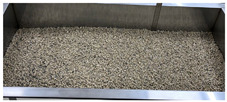

**Table 2 ijerph-19-15666-t002:** Specifications of the main measurement equipment.

Equipment Name	Picture	Specification
Pitot tube	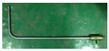	▪ Velocity measurement at the inside of duct▪ Model: 20 cm (Furness)
Micro manometer	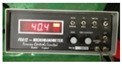	▪ Maximum range: 1.999 kPa▪ Model: FC012 (Furness)
Particle mattersensor	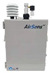	▪ Measurement Range: 0.25 μm~10.0 μm▪ Model PS-1601PMe (HCTM., Republic of Korea)
Air blower(Inverter)	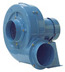	▪ Maximum Range: 20 m^3^/min, 170 mmAq▪ Model DB-270S (Dong-gun, Republic of Korea)
Soil moisturemeter	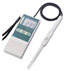	▪ Measurement range: 12.1~58%▪ Model DM-18 (Takemura, Japan)

**Table 3 ijerph-19-15666-t003:** Specifications of the air suction holes of an air sampling port.

Inflow Air Velocity, cm/s	Air Suction Hole Diameter, mm
2	4.1
5	1.8
8	1.4

**Table 4 ijerph-19-15666-t004:** Average value of PM_2.5_ reduction rate for the soil height and the inflow air velocity.

Soil Height (H), mm	Inflow Air Velocity, cm/s		
	2	5	8
100	38.2	35.1	32.5
150	48.7	45.7	43.2
200	57.6	55.3	52.0

**Table 5 ijerph-19-15666-t005:** Ventilation volume according to the inflow air velocities for three active soil filters in classroom.

Inflow Air Velocity, cm/s	Ventilation Volume(A), m^3^/h	Percent (A/Required Volume), %
2	92.3	22.3
5	230.9	55.6
8	369.4	89.0

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
