# Peer review of "Active Soil Filter System for Indoor Air Purification in School Classrooms"

_ijerph, 2022, doi:10.3390/ijerph192315666_

Round 1
Reviewer 1 Report
An interesting, well-structured and readable paper relating to a novel area of research. There are some very minor errors in places, e.g. "have" on line 40 should be "has." Categories (b) and (c) on figure 2 should possibly be switched around? It may be useful to include reference(s) for the statement on lines 243-244 regarding soil moisture levels for crop growth. Otherwise, I would recommend publication of the paper in its current form.
Author Response
Please find the attachment file of the authors reply

Reviewer 2 Report
The reviewed manuscript is primarily descriptive and is at a low level when it comes to the originality of the solutions presented and the detail of the research carried out as well as the final conclusions formulated.
Despite the above reservations, the manuscript brings valuable comments to the discussion and indicates directions for further research related to the problems of active soil filter systems for air purification in school rooms.
However, the presented manuscript contains only a description and results of measurements of the soil filter for air purification for the specific case analyzed here. The description of the conducted studies is clear and does not raise substantive objections, but experimental studies were carried out using simple and standard measuring devices in real conditions of the school room, and the results of these experimental studies are valid only for the specific soil deposit studied. There are no generalizations so that the results can be used for other soil deposits or with a different geometric configuration.
The manuscript lacks any mathematical models and numerical simulations of the process of air filtration through the soil bed, or studies of the harmful impact of PM2.5 concentrations on the environment in school rooms, so that the research has a global and utilitarian dimension. Conclusions from experimental studies of a qualitative nature are obvious and can be formulated without conducting research (e.g. increasing the inflow air velocity through the soil and the stacking height of the multi-layered composite soil in the classroom increases the PM reduction efficiency and pressure loss), conclusions of a quantitative nature apply only to this tested composite soil.
Author Response
Please find the attachment file of the authors reply.

Reviewer 3 Report
This paper tries to explore the effect of active soil filter system on indoor air purification in school classrooms, and the idea is interesting. However, there are some confusing descriptions in this paper, and the authors need to convey the information more clearly and in detail. In addition, it also needs to improve the description of research gap to prove the necessity of this paper. In the whole, this paper can be considered for acceptance after major revision.
1. In Page 1, lines 41-43, “Recently, vegetation filters using plant leaves to purify indoor air have been considered to reduce the PM concentration of indoor pollutants”. This sentence means that the paragraph will be centered on indoor pollution, but the author refers to outdoor pollution later on, such as "vegetation in a park" and "roadside". Context inconsistencies need to be revised.
2. In Page 2, lines 64-65, “but few studies have considered environmental variables, such as airflow velocity through the filter”. Firstly, airflow velocity is not an environmental factor. Factors such as air humidity and temperature are environmental factors. Secondly, I don't understand how the research gap here is logically related to the research content of this paper. Authors need to rewrite research gap.
3. In introduction, the advantages of active soil filter system need to be added, why not use traditional type of filter system for indoor air purification?
4. In Section 2.1, both “415 CMH” and “830 CMH” were mentioned, so in the end which air volume was chosen? Please explain it.
5. In Page 5, lines 131-132, “In the present study, air purification effects were evaluated only for composite soil, excluding plants”. As a passive filtration system, plants inevitably purify the air. How this paper evaluates an air purification system that does not contain plants. Please explain it.
6. In Page 7, lines 178-179, “The PM reduction performance of the soil filter was evaluated by measuring the PM sensor upstream and downstream of the soil filter”. Because the title of this article is “Active Soil Filter System for Indoor Air Purification in School Classrooms”, why only the PM concentration upstream and downstream of the filter was tested instead of the PM concentration in the classroom. Please explain it.
7. In Section 3.1, the reason why the large soil humidity corresponds to large pressure loss needs to be added.
8. In Page 10, lines 236-238, “It is noted that the PM2.5 reduction rate increased as the soil height increased and decreased linearly as the inflow air velocity increased, regardless of the soil height”. The authors had mentioned “the PM2.5 reduction rate increased as the soil height increased” and “regardless of the soil height”, which is contradictory. Please explain it.
9. In Page 12, lines 271-273, “…the energy consumption due to the air suction blower, it is desirable to maintain 30-40% soil moisture and minimize the inflow air velocity to 2 cm/s”. The conclusion here is irresponsible. From the perspective of energy consumption, it is obvious that the smaller the filtration velocity, the lower the energy consumption. The inflow air velocity of 1cm/s will be better than the of 2 cm/s, and the optimal operating conditions given here are meaningless.
10. The title of section 4.3 is “Air Purification Effect of the Active Soil Filters in School Classrooms”, but its research content is almost irrelevant to “air purification effect”. Authors must reorganize this section.
11. In Page 14, lines 328-329, “it not only continuously lowers the PM2.5 concentration inside the classroom but also indirectly controls the indoor humidity during the winter season”. It is mentioned that the soil filter can control the indoor air humidity in winter, but this content has never been involved in the previous paper. I don't understand how the authors come to this conclusion. Please explain it.
Author Response

(The authors gave the same response as above.)

Round 2
Reviewer 2 Report
I would like to thank the authors of the manuscript for their responses to my review and for the corrections and additions made to its content.
The manuscript has been corrected and in this form is eligible for publication, but it is still at a low scientific level. This does not bring anything new to the presented area of research, and a significant improvement would require expanding the scope of experimental research using more advanced measurement techniques and creating an appropriate, more advanced theoretical analysis, reinforced, for example, by numerical simulation of filtration processes for the studied soil structures and operating conditions.
Reviewer 3 Report
The author has made corresponding revisions according to the review comments, and it is recommended that this paper be accepted.